# A solution-to-solid conversion chemistry enables ultrafast-charging and long-lived molten salt aluminium batteries

Jiashen Meng[1], Xuhui Yao[2], Xufeng Hong[1], Lujun Zhu[1], Zhitong Xiao[1], Yongfeng Jia[1], Fang Liu[1], Huimin Song[1], Yunlong Zhao [3] & Quanquan Pang [1] ✉

Conventional solid-to-solid conversion-type cathodes in batteries suffer from poor diffusion/reaction kinetics, large volume changes and aggressive structural degradation, particularly for rechargeable aluminium batteries (RABs). Here we report a class of high-capacity redox couples featuring a solution-to-solid conversion chemistry with well-manipulated solubility as cathodes—uniquely allowed by using molten salt electrolytes—that enable fast-charging and long-lived RABs. As a proof-of-concept, we demonstrate a highly reversible redox couple—the highly soluble InCl and the sparingly soluble $InCl_3$—that exhibits a high capacity of about 327 mAh g$^{-1}$ with negligible cell overpotential of only 35 mV at 1 C rate and 150 °C. The cells show almost no capacity fade over 500 cycles at a 20 C charging rate and can sustain 100 mAh g$^{-1}$ at 50 C. The fast oxidation kinetics of the solution phase upon initiating the charge enables the cell with ultrafast charging capability, whereas the structure self-healing via re-forming the solution phase at the end of discharge endows the long-term cycling stability. This solution-to-solid mechanism will unlock more multivalent battery cathodes that are attractive in cost but plagued by poor reaction kinetics and short cycle life.

Confronting the threat of climate change and energy crisis, grid-scale energy storage system (ESS) is recognized as the essential building block for promoting sustainable energy supply in the market[1,2]. Although lithium-ion batteries (LIBs) are dominating the electrochemical sector of current ESS market, the scarcity and high cost of lithium resources greatly restrict their penetration, more so given the high demand from the electric vehicle sector[3,4]. The field has to identify promising alternatives to LIBs, and for such, rechargeable aluminium batteries (RABs) have emerged as a promising candidate due to the low cost, air-exposure safety and high abundance of aluminium (8.1 wt% in the Earth's crust), and yet the aluminium metal anode has a rather high theoretical capacity (8040 mAh cm$^{-3}$), which significantly contributes to the development of high-capacity RABs[5-7].

However, the development of viable RABs faces tough challenges including limited charge storage capacity, poor reaction kinetics or very short cycling life[8,9]. The efforts in designing high-performance RABs can be traced by the underlying charge-compensation mechanism[10-13]. Considerable efforts started with cells using graphite-based cathodes and room temperature chloroaluminate ionic liquid electrolytes that operate by fast insertion of chloroaluminate anions ($AlCl_4^-$) into graphite layers; the cathodes exhibited a wide range of capacity between 60 ~ 200 mAh g$^{-1}$ and have shown applications in large-scale cells[14-17]. Notably, some materials have been developed to insert $Al^{3+}$ with higher capacities (100-300 mAh g$^{-1}$)—such as $TiO_2$, $Mo_6S_8$ and vanadium-MXene—but suffer from large overpotential (0.4–0.8 V) and poor rate capability. These limitations

[1]Beijing Key Laboratory of Theory and Technology for Advanced Batteries Materials, School of Materials Science and Engineering, Peking University, Beijing 100871, China. [2]Advanced Technology Institute, Department of Electrical and Electronic Engineering, University of Surrey, Guildford GU2 7XH, UK. [3]Dyson School of Design Engineering, Imperial College London, London SW7 2BX, UK. ✉e-mail: qqpang@pku.edu.cn

are mainly attributed to the strong electrostatic repulsive force from material lattices to $Al^{3+}$ carriers[18–20].

Unlike the insertion-type mechanism, materials based on conversion-type chemistry express outstanding capacities by invoking phase change with complete structure reconstruction. As universally applied to lithium-ion chemistry, the typical solid-to-solid conversion chemistry suffers from poor diffusion/reaction kinetics, large volume changes and electrode structural deterioration[21]. The solution-to-solution chemistry, wherein the two end products are both largely solvated such as in the redox flow batteries, albeit free of structural degradation, involves substantial active material diffusion and cross-reaction with the anode causing capacity fade and low Coulombic efficiency (CE), unless a highly delicate ion-selective membrane is implemented[22]. For RABs, most reports utilized transition metal chalcogenides, which are based on the solid-to-solid conversion chemistry and exhibit capacities up to ~400 mAh g⁻¹ based on solid-to-solid conversion to the elemental metal (or its chalcogenides)[23–26]. Although the high capacity has attracted increasing research interest of battery community, the underlying conversion reaction mechanism remains ambiguous. There are still many unresolved problems in the reaction, including high polarization, low rate capability and severe capacity fading at early stage. These problems are even severe in multivalent aluminium chemistry than in the monovalent counterparts because of the sluggish nature of multivalent carriers[5].

In this work, we report a class of high-capacity redox couples featuring solution-to-solid conversion chemistry with well-manipulated solubility as cathodes—uniquely enabled by using molten salt electrolyte—that endows RABs with fast charging capability and long lifetime. Unlike the conventional solid-to-solid conversion reactions, this solution-to-solid reaction features intrinsic fast kinetics and structural self-healing characteristics. The success of such cathodes is demonstrated by a highly reversible redox couple converted between soluble InCl (In⁺) and sparingly soluble $InCl_3$ (In³⁺) via a two-electron-transfer process (schematically shown in Fig. 1). The benefits from the solution-phase InCl are two-fold—the rapid solution phase oxidation allows ultrafast charging ability, and its reformation in each discharging process endows structure self-healing and high capacity retention. It is worth noting that the soluble phase is not the oxidized product, by which its cross-reacting with the anode is prevented. The inorganic chloroaluminate molten salt electrolyte used in this work not only permits the aluminium cells to operate at moderately elevated temperatures (130–150 °C), but also allows manipulation of the solubility of the two end products.

## Results

### The indium electrochemistry in the molten salt electrolytes

The eutectic mixture of $AlCl_3$-NaCl-KCl exhibits a low melting point of around 95 °C due to the unique covalent bonding within $Al_nCl_{3n+1}^-$ moieties[27,28]. It also provides high desolvation kinetics at moderate temperatures, unlike the liquid metal or sodium-sulfur batteries that typically require 250–600 °C. We first investigate the indium-based solution-to-solid chemistry using a model configuration with an InCl saturated electrolyte as the source of active cathode material. The indium metal spontaneously reduces the chloroaluminates to generate monovalent In⁺ (and forming metal aluminium) that is highly soluble in the chloroaluminate melt (In⁺/In redox potential is lower than $Al^{3+}$/ Al)[29,30]. This reaction was validated via the generation of aluminium metal on the surface of indium foil in the chloroaluminate melt, detected by X-ray diffraction (Fig. 2a) and scanning electron microscopy (SEM) imaging (Supplementary Fig. 1). The dynamically generated monovalent In⁺ converts to trivalent indium(III) upon charge, $InCl_3$, which is only sparingly soluble in the chloroaluminate melt. The manipulation of In⁺/$InCl_3$ solubility is unique to the molten salt as we observe that the In⁺ shows very low solubility in the organic ionic liquid counterpart and very poor reaction kinetics, as discussed below.

To understand the nature of the indium electrochemistry in chloroaluminate melt electrolyte, a three-electrode beaker-type cell (Fig. 2b) was fabricated using aluminium as both counter and reference electrodes, molybdenum as the working electrode, and an indium foil immersed in the chloroaluminate melt to supply InCl. The cyclic voltammetry (CV) measurement (Fig. 2c) at 110 °C shows a sharp anodic peak at 1.15 V and a broad cathodic peak at 0.68 V, indicating an asymmetric reaction. It is worth noting that the temperature has a great impact on the overpotential of cells and the solubility of indium in the chloroaluminate melt, which is revealed by the temperature-variant CV (Fig. 2c) and inductively coupled plasma optical emission spectrometry (indium: 1.24 wt% at 110 °C and 5.05 wt% at 190 °C) (Fig. 2d). The solubility here is significant as the dissolution of In⁺ is a dynamic process which continues as its oxidation occurs. In contrast, the $InCl_3$ shows a negligible indium concentration of 0.02 wt% at 150 °C in the chloroaluminate melt, confirming its sparingly soluble property (Supplementary Table 1).

We further demonstrate the reversibility of such chemistry via an Al||Mo cell with a home-made Swagelok configuration (Fig. 2e). An indium foil was sandwiched between two glass fiber separators to supply InCl (cell denoted as Al|Mo/In). Theoretically, the temperature of the Al||Mo/In cell ought to fall between the melting point of indium metal (~156 °C) and of the chloroaluminate melt (~95 °C). Here, three

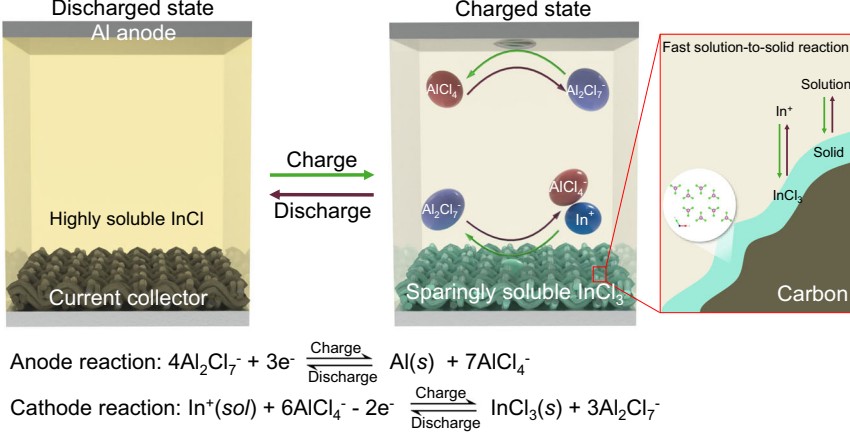

**Discharged state** — Al anode — Highly soluble InCl — Current collector

**Charged state** — $AlCl_4^-$, $Al_2Cl_7^-$, $Al_2Cl_7^-$, $AlCl_4^-$, In⁺ — Sparingly soluble $InCl_3$ — Fast solution-to-solid reaction — Solution — In⁺ — $InCl_3$ — Solid — Carbon

Charge / Discharge

Anode reaction: $4Al_2Cl_7^- + 3e^- \xrightleftharpoons[\text{Discharge}]{\text{Charge}} Al(s) + 7AlCl_4^-$

Cathode reaction: $In^+(sol) + 6AlCl_4^- - 2e^- \xrightleftharpoons[\text{Discharge}]{\text{Charge}} InCl_3(s) + 3Al_2Cl_7^-$

**Fig. 1 | Schematic illustration of the proposed solution-to-solid indium conversion chemistry and the rechargeable aluminium battery.** The reduced In⁺ (as in InCl) is highly soluble in the chloroaluminate molten salt electrolyte, whereas the oxidized $InCl_3$ is sparingly soluble. Upon charge, the solution-phase In⁺ is oxidized to solid $InCl_3$ based on a surface-precipitation reaction; and upon discharge, the sparingly soluble $InCl_3$ converts to soluble In⁺, featuring structural self-healing.

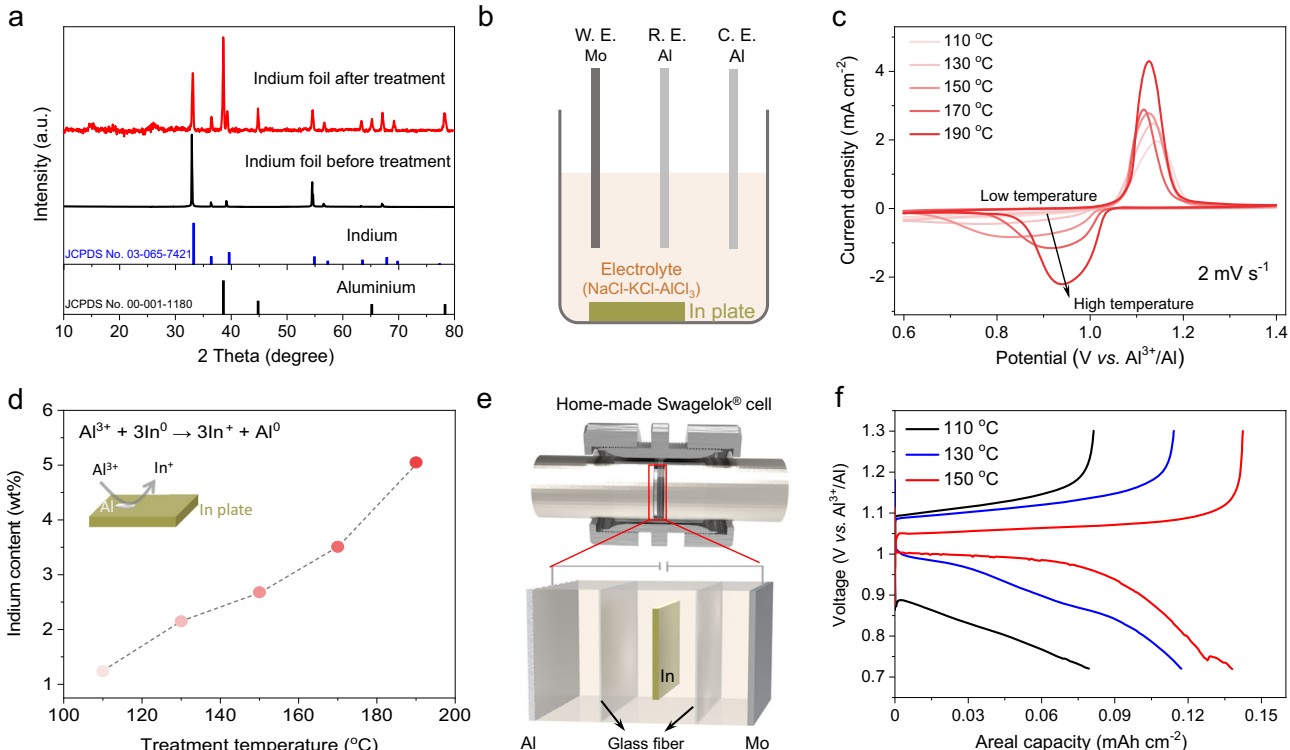

**Fig. 2 | Electrochemical characterization of the indium conversion electrochemistry in the molten salt electrolyte. a** XRD patterns of the indium foil before and after treatment in the chloroaluminate melt at 190 °C. **b** Schematic illustration of the three-electrode cell with an indium foil immersed in the electrolyte, a Mo wire as the working electrode (W.E.) and an Al wire as the counter electrode (C.E.) and reference electrode (R.E.). **c** CV plots of the three-electrode cell at different operation temperatures (scan rate of 2 mV s⁻¹). **d** The concentrations of indium element in the chloroaluminate melt as measured by inductively coupled plasma optical emission spectrometry. **e** Schematic illustration of the Al|Mo/In battery configured with an indium foil between two separators in a home-made Swagelok cell. **f** The voltage profiles of the Al|Mo/In battery at different operation temperatures (0.5 mA cm⁻²).

different attempts at different temperatures of 110 °C, 130 °C, and 150 °C were made with a current density of 0.5 mA cm⁻². The Al|Mo/In cells showed overpotentials of 0.16, 0.10 and 0.04 V, respectively, and high CE of above 95% (Fig. 2f). One must note that the indium metal cannot physically contact the cathode; otherwise, the spontaneously formed metal aluminium would result in "zero" open circuit voltage of the Al|Mo/In cell (Supplementary Fig. 2). In addition, to prove that the deposit on the anode side is pure aluminium and indium-free, electroplating process was conducted on a Mo foil using the Al|Mo/In cell. The elemental analysis shows that, no indium signal was observed on the Al deposit, confirming that the deposit is exclusively Al metal (Supplementary Fig. 3).

As the solution-to-solid conversion occurs via a surface-precipitation path, the electrode surface area is vital for the areal capacity of the cell. Carbon cloth (CC) was thus chosen as a free-standing electrode to evaluate the capacity performance of such chemistry in the cell. A controlled calcination process was performed on the CC to increase the surface area from 1.1 m² g⁻¹ to 375.8 m² g⁻¹ (activated carbon cloth, denoted as ACC) (Supplementary Fig. 4)[31]. One must note here that the ACC has a low degree of graphitization so that the insertion contribution of AlCl₄⁻ upon charge can be negligible.

Three types of cells, Al|Mo/In, Al|CC/In, and Al|ACC/In, with an excess supply of InCl in solution, were assembled and evaluated at 150 °C. As shown in Fig. 3a, the specific areal discharge capacities of these cells at a current density of 0.5 mA cm⁻² are 0.13, 2.2, and 3.2 mAh cm⁻², respectively. These results show that the capacity is limited by the surface area, indicating that a surface precipitation process should be involved in the reaction (the same as the gas-to-solid case in metal-air batteries) (Supplementary Fig. 5a). The good reversibility of the solution-to-solid conversion is confirmed via well-defined anodic/cathodic peaks at 1.18 and 0.89 V by CV measurements (Fig. 3b). The charging reaction kinetics and CE of the cell were evaluated by cycling under varied charging current densities (fixed discharge current) and a charge capacity cut-off of 2 mAh cm⁻² (as InCl is dynamically supplied and unlimited) (Fig. 3c). When charged at a current density of 2 mA cm⁻², the cell shows a small overpotential of 0.11 V with CE of ~98.6%. At higher charging rates of 5, 10, and 20 mA cm⁻², the cell still delivers average discharge capacities of 1.25, 0.92, and 0.76 mAh cm⁻², respectively. The well-defined plateau behavior is maintained at each charging rate (up to 20 mA cm⁻²), as shown in the voltage profiles (Fig. 3d). In addition, the discharging rate performance of the Al|ACC/In cell was evaluated (Supplementary Fig. 5b, c). The cell retains rather high areal discharge capacities of 1.93, 1.66, and 1.42 mAh cm⁻² with low voltage polarization at low current densities of 1, 2 and 3 mA cm⁻², respectively. At higher rates of 5 and 10 mA cm⁻², the cell shows substantial capacities of 1.12 and 0.84 mAh cm⁻², although the voltage polarization is higher. Therefore, the cell shows an inferior discharging rate performance than the charging, but is nevertheless satisfactory for practical applications particularly compared with the conventional conversion reaction-based aluminium batteries. This is fundamentally owed to the use of chloroaluminate melt electrolyte that shows very high desolvation kinetics, and to the presence of dissolved InCl as the discharge product. It is noted that the overpotential on charge is higher than that on discharge at the same rates, again confirming asymmetric reactions. The cycling performance of the Al|ACC/In cell (Fig. 3e) shows stable cycling over 100 cycles with 2 mAh cm⁻² capacity at a current density of 1 mA cm⁻². Even at a high charging rate of 10 mA cm⁻², the Al|ACC/In cell still delivers high capacity retention of 90.6% for over 100 cycles (Fig. 3f).

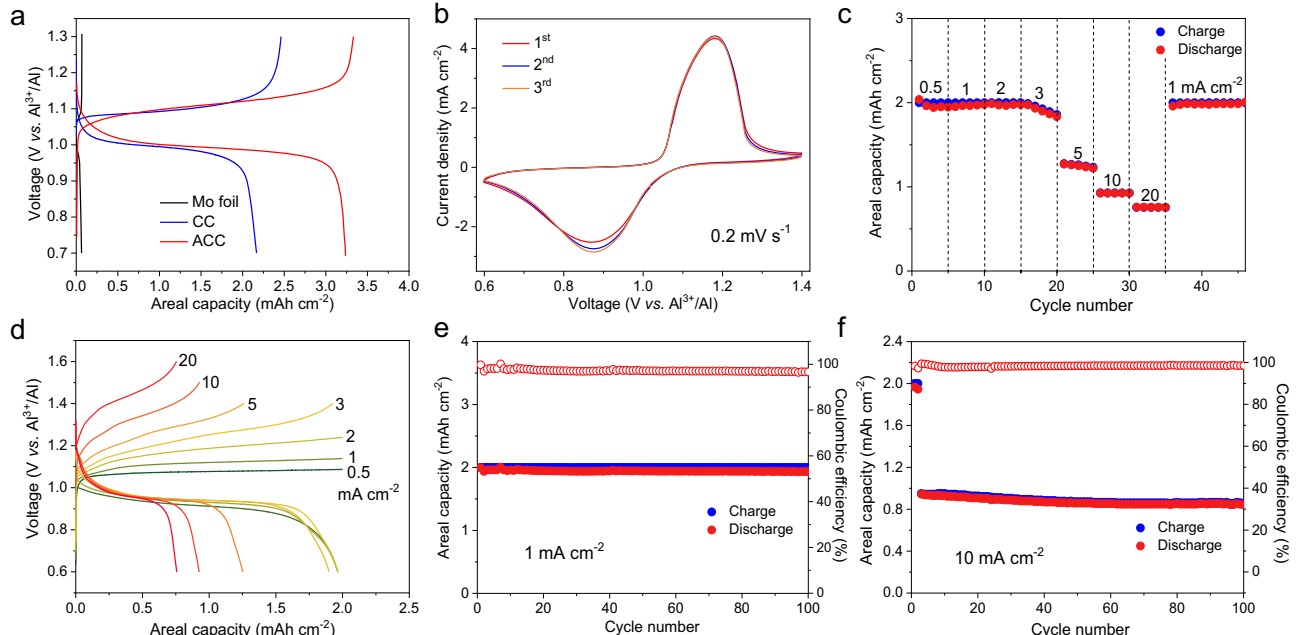

**Fig. 3 | Electrochemical performance of the solution-to-solid conversion electrochemistry of the Al | ACC/In cells. a** The voltage profiles of cells using varied electrodes at 0.5 mA cm⁻². **b** CV curves of the Al|ACC/In cell at a scan rate of 0.2 mV s⁻¹. **c** Rate performance of the Al|ACC/In battery at different charging rates ranging from 0.5 to 20 mA cm⁻² with a fixed discharging rate of 0.5 mA cm⁻² (charge capacity cut-off is 2 mAh cm⁻²), and (**d**) the corresponding voltage profiles. **e** Cycling performance of the Al|ACC/In cell charged at 1 mA cm⁻² and discharged at 0.5 mA cm⁻² with a charge capacity cut-off of 2 mAh cm⁻². **f** Cycling performance of the Al|ACC/In cell at a high charging current of 10 mA cm⁻² and a discharging rate of 0.5 mA cm⁻².

As a demonstration of high mass loading, Al|ACC/In cells with two and three layers of ACC electrodes were assembled, and the cells show higher areal capacities of 5.75 and 8.77 mAh cm⁻² (Supplementary Fig. 5d). These results indicate that the solution-to-solid chemistry is not kinetically limited by electrode-scale mass transfer or the aluminium plating/stripping kinetics at the anode, which is attributed to the excellent kinetics of the chloroaluminate melt electrolyte. For reference, a cell assembled with an organic ionic liquid electrolyte showed a very low capacity of ~0.09 mAh cm⁻² and low CE of below 50% at 150 °C (Supplementary Fig. 5e, f). This can be attributed to negligible formation of soluble InCl, evidenced by a very low indium concentration of 0.17 wt% in the ionic liquid electrolyte (Supplementary Table 1). The low CE can be attributed to a certain solubility of the generated InCl₃ in hot ionic liquid electrolyte, leading to serious shuttle effect during charging. This obvious difference confirms the unique physicochemical property of indium chemistry enabled by the chloroaluminate melt.

**The cathode conversion mechanism**

The reaction mechanism of reversible indium-based solution-to-solid conversion is proposed as follows. The indium metal in the melt serves to supply the highly soluble InCl (Eq. 1), and the active chloroaluminate complexes contain $AlCl_4^-$ and $Al_2Cl_7^-$ and other long-chain $Al_3Cl_{10}^-$ species[28]. The reaction with indium metal requires Al-Cl bonds breaking and therefore, the species that has weak Al-Cl bonding would preferentially react with the indium metal. Therefore, the longer-chain $Al_3Cl_{10}^-$ and $Al_2Cl_7^-$ species have weaker Al-Cl bridging bonds than the $AlCl_4^-$ species, which preferentially reacts with indium. In addition, these chloroaluminate complexes serve as the Cl⁻ mediator to facilitate the solution-to-solid conversion between InCl and InCl₃ (Eq. 2). Upon charge, the highly soluble In⁺ is rapidly oxidized to sparingly soluble InCl₃ (along with the conversion from $AlCl_4^-$ to $Al_nCl_{3n+1}^-$); upon discharge, the sparingly soluble InCl₃ converts back to InCl, recovering the solution state and eliminating the structural degradation that plagues solid-to-solid conversion. On the anode side, Al plating and stripping occur with the $AlCl_4^-/Al_nCl_{3n+1}^-$ conversion (Eq. 3)[32]. The overall reaction is shown in Eq. 4.

InCl generating reaction:

$$Al_nCl_{3n+1}^- + 3(n-1)In(s) \rightarrow 3(n-1)InCl(sol) \\ + (n-1)Al(s) + AlCl_4^- (n \geq 2) \tag{1}$$

Cathode reaction:

$$(n-1)In^+(sol) + 3nAlCl_4^- - 2(n-1)e^- \leftrightarrow (n-1)InCl_3(s) + 3Al_nCl_{3n+1}^- \tag{2}$$

Anode reaction:

$$4Al_nCl_{3n+1}^- + 3(n-1)e^- \leftrightarrow (n-1)Al(s) + (3n+1)AlCl_4^- \tag{3}$$

Overall reaction:

$$3(n-1)In^+(sol) + (3n-2)AlCl_4^- \leftrightarrow 3(n-1)InCl_3(s) \\ + 2(n-1)Al(s) + Al_nCl_{3n+1}^- \tag{4}$$

To investigate the structural and chemical evolution of the cathode during the electrochemical process, ex-situ XRD patterns of the ACC cathode at different states were studied (Fig. 4a). At 50% SOC, two peaks corresponding to the (200) and ($\bar{4}$02) planes of InCl₃ appear (JCPDS no. 034-1145) and increase upon further charge, indicating the formation of solid phase InCl₃ on the cathode surface. The intensity of InCl₃ peaks weakens at 50% DOD and completely vanishes in the fully discharged state, showing excellent reversibility from the solid phase to the solution. X-ray photoelectron spectroscopy (XPS) was used to track the coordination state change of indium during the electrochemical process (Fig. 4b and Supplementary Fig. 6). The In $3d$ spectrum before cycling shows two peaks centered at 452.8 and 445.2 eV, corresponding to the monovalent In⁺ and the splits of In $3d_{3/2}$ and In $3d_{5/2}$ core levels[33]. After fully charged, the two peaks shift to higher

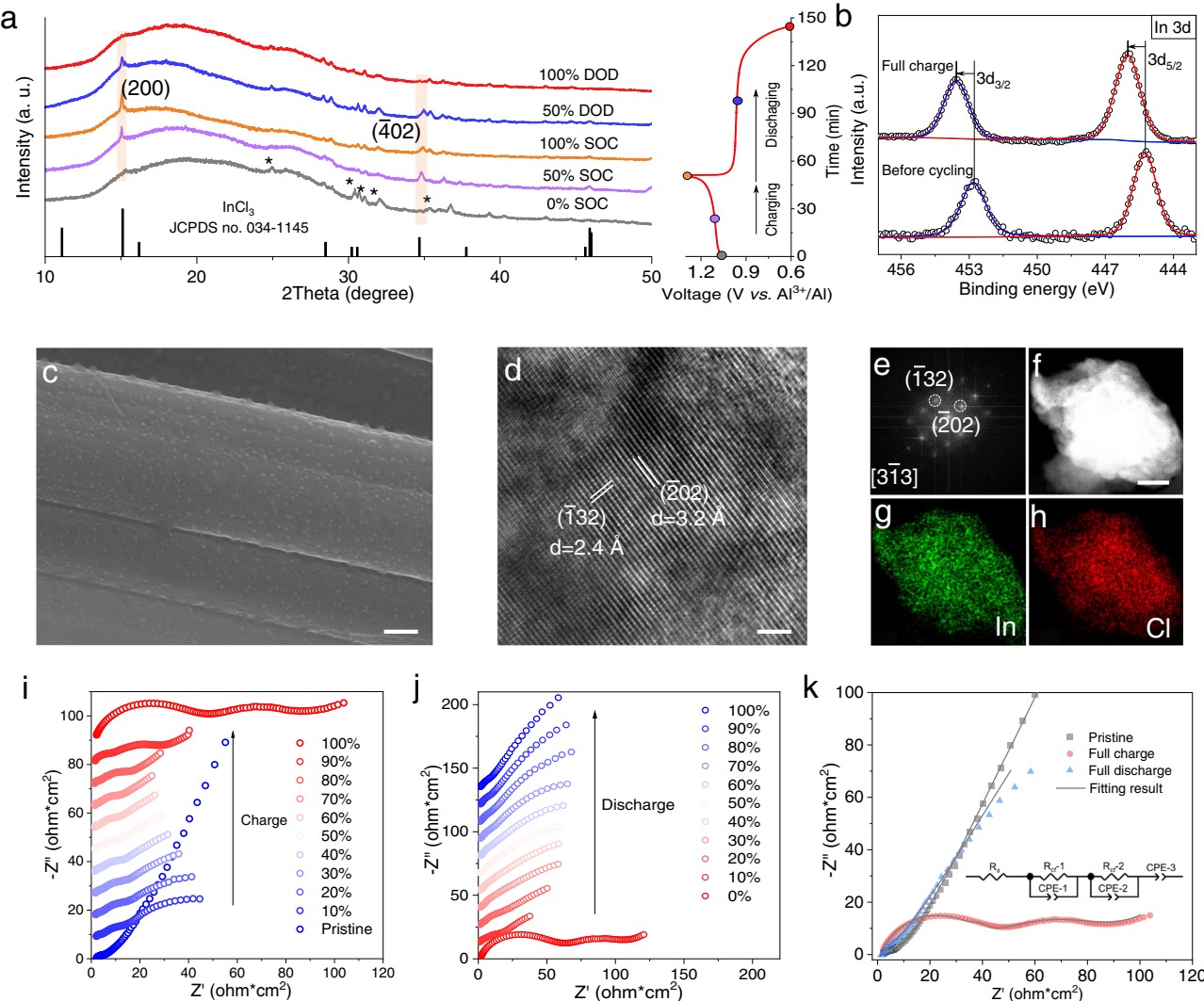

**Fig. 4 | Understanding of the solution-to-solid chemistry mechanism in the Al | ACC/In cell. a** Ex-situ XRD patterns at different states. A discharging rate is half of a charging one to ensure full reaction on discharge. The weak diffraction peaks marked by asterisks belong to the residual chloroaluminate melt electrolyte. **b** The XPS spectra of ACC cathode in Al|ACC/In battery before cycling and after full charge. **c** The SEM image of the ACC cathode after full charge, scale bar: 1 μm. **d**, **e** TEM images and FFT pattern of the fully charged product; **d** scale bar: 2 nm. **f–h** The HAADF-STEM image and the corresponding element mapping images of the fully charged product; scale bar: 200 nm. **i**, **j** In situ time-lapse EIS profiles of the Al|ACC/In cell at different states of charge (SOC) and depths of discharge (DOD) during the initial cycle. **k** EIS spectra measured at the pristine state, after full charge and after full discharge (inset is the equivalent circuit model for the pristine state and full discharge). HAADF-STEM, high-angle annular dark-field scanning-TEM.

binding energy, showing higher-valence InCl₃[33]. The SEM images of the cathode in the fullly charged state show a rough surface covered by abundant solid nanoparticles with uniformly distributed In/Cl elements (Fig. 4c and Supplementary Fig. 7a–c). The transmission electron microscopy (TEM) image shows two ordered lattice fringes with spacing distances of about 2.4 Å and 3.2 Å, corresponding to the ($\bar{1}$32) and ($\bar{2}$02) planes of InCl₃ (Fig. 4d). The fast Fourier transformation (FFT) pattern indicates an InCl₃ phase along the [3$\bar{1}$3] zone axis (Fig. 4e), and element mappings confirm uniform distributions of In and Cl (Fig. 4f–h and Supplementary Fig. 7d–f).

In-situ EIS measurement was performed to study the kinetics of the solution-to-solid conversion (Fig. 4i, j). The pristine Al|ACC/In cell shows a very small solution resistance ($R_s$) of 1.8 Ω and interface charge-transfer resistance ($R_{ct}$) of 8.8 Ω. Upon charge (SOC of 10%), a new semicircle at the low-frequency region appears, which is attributed to the formation of the solid-phase InCl₃ creating a new interface. This is further supported by high-accuracy distribution of relaxation time (DRT) analysis which mathematically converts the frequency-dependent EIS spectra to relaxation-based functions γ(τ) (details in the Method section)[34,35]. At 10% SOC, three typical peaks located at about 0.07 s, 0.65 s, and 5.6 s are observed, corresponding to the $R_{ct}$ of the Al anode, the cathode, and a new interphase, respectively[36], based on the frequency responses of different interfacial processes (Supplementary Fig. 8). During subsequent charge (SOC: 10% to 80%), the $R_{ct}$ values remain constant at roughly 6.4 Ω·cm², indicating fast and unaffected conversion kinetics from a solution to solid (Supplementary Fig. 8b). Nonetheless, the shifting DRT peaks to high value indicate the accumulation of solid InCl₃ (Supplementary Fig. 8b, c). Beyond SOC of 90%, the $R_{ct}$ starts to increase, because the precipitation of InCl₃ leads to a decrease of the available surface sites for reaction, until two large semicircles appear at 100% SOC indicating complete surface covering. Upon discharge, the $R_{ct}$ rapidly decreases because of the reductive dissolution of solid InCl₃ (Fig. 4j and Supplementary Fig. 8d). Subsequently, the slope of the low-frequency lines gradually increases, indicating the increasing ability for mass transfer (of soluble InCl)[37]. In short, the direct comparison of the EIS spectra at pristine, fully charged

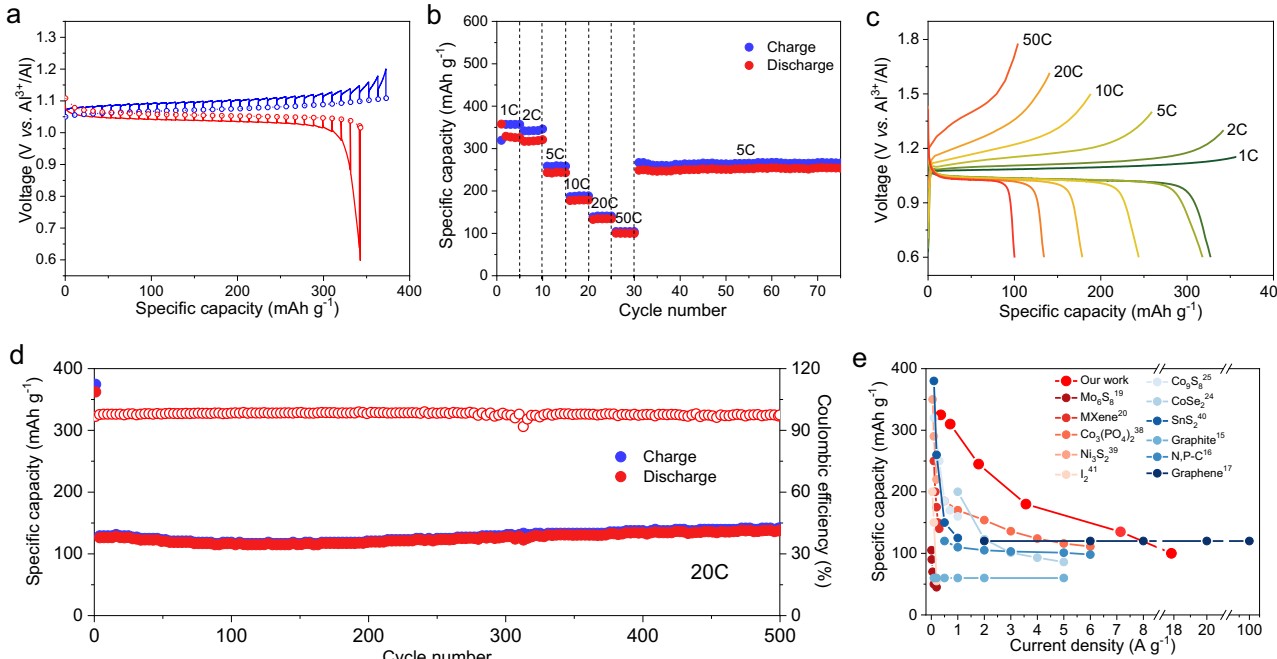

**Fig. 5 | Electrochemical performance of the Al|ACC/InCl cell based on the solution-to-solid conversion chemistry. a** The GITT voltage profile of the Al|ACC/InCl cell at a rate of 0.2 C. **b** The charging rate performance of the Al|ACC/InCl cell at varied charging rates from 1 C to 50 C with a fixed discharging rate of 0.5 C, and (**c**) the corresponding voltage profiles. **d** Cycling performance of the cell with a charging rate of 20 C and a discharging rate of 0.5 C. **e** Comparison of the electrochemical performances demonstrated in our work and previous reports.

and discharged states (Fig. 4k and Supplementary Fig. 8e) proves the oxidation of a majorly soluble phase (InCl) on charge, and the reduction of a sparingly soluble phase (InCl₃) on discharge. Note that the $R_s$ value varies negligibly showing unaffected mass transport in the chloroaluminate melt (Supplementary Fig. 8f).

## Cells directly using InCl cathode

In order to realize this solution-to-solid conversion chemistry on the basis of conventional battery production infrastructure, cells were fabricated using powder InCl cathodes directly (although InCl can be also added into the electrolyte). In the Al|ACC/In cell shown in the above section, an indium metal was used in the cell to dynamically supply dissolved In(I), and the In(I) can be considered in excess and the available carbon surface is the limited factor. Therefore, the "areal" matrice is used to show the fundamental electrochemical behavior of the In(I)/In(III) redox couple. In this part, powder InCl cathode was used to fit in the evaluation framework of a standard battery, so that the mass of InCl is limited while the carbon surface is potentially in excess. In brief, this difference between the two cells on current density and capacity depends on which one is the limiting factor for capacity.

The theoretical gravimetric capacity of InCl is calculated to be 357 mAh g⁻¹, and 1 C corresponds to a current density of 357 mA g⁻¹. The galvanostatic intermittent titration technique (GITT) profile (Fig. 5a) shows a quasi-equilibrium voltage of ~1.05 V, and a discharge capacity of 343 mAh g⁻¹ at 0.2 C that is very close to theoretical value. The self-discharge property of the Al|ACC/InCl was evaluated by resting upon full charging (Supplementary Fig. 9, further discussions in Supplementary Fig. 9). A high charge retention of ~95.2% indicates relatively low self-discharge property at a high operation temperature, attributed to the low solubility of solid InCl₃ in the molten salt electrolyte.

The fast-charging performance was evaluated by charging at different current rates from 1 C to 50 C but with a fixed discharging rate of 0.5 C. The cells deliver excellent capacities of 327, 319, 244, and 179

mAh g⁻¹ at 1 C, 2 C, 5 C, and 10 C, respectively (Fig. 5b). It is worth noting that capacities of 134 and 100 mAh g⁻¹ are achieved at fast charging rates of 20 C and 50 C, respectively. Furthermore, the voltage profiles (Fig. 5c) show well-defined voltage plateaux and low overpotential, which favors high round-trip energy efficiency (EE) that is critical for ESS applications (for example, 0.11 V overpotential and ~83% EE at 10 C). The cell exhibits excellent cycling stability at 5 C and 10 C, with almost unvaried voltage profiles over cycling (Supplementary Fig. 10). At a high charging rate of 20 C, the cell shows excellent cycling stability with no capacity fade after 500 cycles (Fig. 5d). Compared with most of previously reported RABs, our cells display high performance in terms of specific capacity and current density[15–17,19,20,24,25,38–41], which relies on the revolutionary solution-to-solid chemistry (Fig. 5e). In addition, our Al|ACC/InCl cells with high InCl mass loadings still exhibit a high reversible capacity over 200 mAh g⁻¹ at a low rate and good cycling stability, confirming the potential of this solution-to-solid reaction chemistry in practical application (Supplementary Fig. 11, further discussions in Supplementary Fig. 11).

## Outlook on the solution-to-solid conversion cathodes

The solution-to-solid conversion chemistry can be easily extrapolated to other redox pairs for fast-charging and long-lived RABs. Low-valence CuCl, SnCl₂, CrCl₂, and MnCl₂ were also studied as active materials. They all show considerate solubility in the chloroaluminate melt but different electrochemical behavior (Supplementary Fig. 12a). In open-circuit rest, the Al|ACC/CuCl and Al|ACC/SnCl₂ cells show fast voltage decay to 0 V, which could be attributed to the spontaneously reduction reaction of CuCl and SnCl₂ by Al anode. This reaction could result in the plating of dendritic Cu and Sn metals and then lead to short circuit of the cell (Supplementary Fig. 12b, c). In contrast, the Al|ACC/CrCl₂ and Al|ACC/MnCl₂ cells exhibit stable open circuit voltage, because no reduction reaction occurred due to the redox potentials of Cr and Mn close to that of Al³⁺/Al (Supplementary Fig. 12d, e). The Al|ACC/CrCl₂ cell shows a capacity of ~58 mAh g⁻¹ and a well-defined voltage plateau at ~0.9 V with a high CE of 92% (Supplementary

Fig. 12d). For the Al|ACC/MnCl$_2$ cell, severe shuttling is observed, which is attributed to the high solubility of the high-valence MnCl$_4$ in the chloroaluminate melt (Supplementary Fig. 12e). Therefore, we propose two criteria to describe the stability of redox pair in solution-to-solid conversion chemistry as follows. (1) the material in the reduced state (noted as M$^{red+}$) should be highly soluble in the electrolyte, and the M$^{red+}$/M(0) couple should possess a more negative redox potential than the Al(III)/Al(0) couple; (2) the material in the oxidized state (noted as M$^{ox+}$) should be sparingly soluble or insoluble in the electrolyte, and the M$^{ox+}$/M$^{red+}$ couple should have high potential to generate a high cell voltage.

In terms of practical aspects, we demonstrate that a high areal capacity of ~9.2 mAh cm$^{-2}$ and a high CE of above 96% can be achieved using a carbon nanotube film with high surface area as the current collector (Supplementary Fig. 13a). Furthermore, a cell using this chemistry possesses excellent thermal shock stability because the solution-to-solid chemistry is microstructure-insensitive (and self-healing). A freeze-thaw test indicates the Al|ACC/InCl cell returns to normal operation without capacity decay at 150 °C (Supplementary Fig. 13b).

## Discussion

In summary, we reported a successful demonstration of solution-to-solid conversion chemistry in RABs, featuring fast charging and long-lived performance. This solution-to-solid reaction effectively breaks through the sluggish kinetics in solid-to-solid conversion reaction that plagues conventional electrodes. The RABs based on such chemistry with highly reversible InCl and solid InCl$_3$ showed fast charging capability (~100 mAh g$^{-1}$ at 50 C, that is, minute-charge) and very stable cycling with no capacity fade after 500 cycles at a charging rate of 20 C. The chloroaluminate melt electrolyte used in these RABs is vital because it enables the manipulation of the In$^+$/InCl$_3$ solubility while organic ionic liquid cannot, which is necessary to achieve the solution-to-solid chemistry. Besides, the fast Al$^{3+}$ desolvation kinetics of the chloroaluminate melt electrolyte at a moderate temperature significantly contributes to the fast charging capability. The non-flammable inorganic melt ensures the intrinsic safety of batteries, which is attractive for grid-scale applications.

The structure self-healing characteristic of solution-to-solid conversion endows high thermal shock ability, so the cells can endure frequent shutdown and exhibit long shelf life by maintaining the frozen state when not in use. The high energy efficiency (e.g., ~83% at a high charging rate of 10 C) of the cell makes it highly advantageous in serving the peak load regulating of the grid. In addition to the system demonstrated in our work, it is reasonable to believe that more solution-to-solid conversion based multivalent batteries can be developed in the same methodology. Given the intrinsic safety and low cost, together with the durability, fast-charging ability, and high energy efficiency, the battery based on such solution-to-solid conversion chemistry principle would become an affordable large-scale energy storage solution that contributes to realizing a Net-Zero future.

## Methods
### Preparation of the electrolytes
All the preparation procedures below were performed in an argon-filled glovebox (O$_2$ < 0.01 ppm, H$_2$O < 0.01 ppm). The inorganic chloroaluminate melt electrolyte consisted of anhydrous AlCl$_3$ (Acros, 99.9%), KCl (Aladdin, 99.99%) and NaCl (Aladdin, 99.99%). The KCl and NaCl were pre-dried to 500 °C in a muffle furnace before use. Based on the phase diagram, a molar ratio of 61:13:26 (AlCl$_3$, KCl, and NaCl) was used to prepare the electrolyte with an eutectic point[27]. All salts were mixed into a glass weighing flask, and the flask was sealed and heated to 150 °C in an oven for 24 h. The mixture appeared as a homogeneous clear liquid but transformed into a solid after cooling to room temperature. This solid was then ground thoroughly to obtain a powder electrolyte. The organic ionic liquid electrolyte consisted of 1-ethyl-3-methylimidazolium chloride (EMIC, Sigma-Aldrich, 98%) and anhydrous AlCl$_3$ with a molar ratio of 1:1.3. The EMIC was dried at 150 °C for 12 h before use, committing to remove the residual water via a melting and recrystallization process. During the mixing process, the anhydrous AlCl$_3$ was slowly added to the EMIC powder with vigorous stirring at room temperature, forming a clear electrolyte liquid at the end.

### Materials characterizations
XRD characterization was performed using a D8 Advance X-ray diffractometer with a Cu Kα X-ray source. For the ex-situ XRD study, the Al|ACC/In cells (in Swagelok® configuration) were charged at a rate of 3 mA cm$^{-2}$ and discharged at a rate of 1.5 mA cm$^{-2}$ (temperature: 150 °C). The cells were stopped at states of 50% SOC, 100% SOC, 50% DOD, and 100% DOD, respectively. After cooling down, the ACC cathodes were taken out and sealed by two Kapton films in an argon-filled glovebox to avoid air exposure. Raman spectra were collected using a Renishaw INVIA micro-Raman spectroscopy system. XPS measurement was performed using a VG MultiLab 2000 instrument. For the ex-situ XPS study, the Al|ACC/In cells (before and after the charging process) were disassembled in an argon-filled glovebox. The ACC cathodes were taken out and transferred into the high vacuum XPS instrument chamber via a joint glove box in order to avoid air/moisture exposure. SEM images were captured using a JEOL JSM-7100F instrument with an acceleration voltage of 20 kV. Elemental mapping was conducted using an EDX-GENESIS 60 S spectrometer. The surface area and pore size distribution were measured based on nitrogen adsorption isotherm performed at 77 K using a Tristar-3020 instrument, and the Brunauer-Emmett-Teller (BET) method was used to calculate the surface area and Barrett-Joyner-Halenda (BJH) method was used for pore size distribution calculation. Transmission electron microscopy (TEM), scanning transmission electron microscopy (STEM) and energy dispersive X-ray spectroscopy (EDS) mapping images were obtained on a ThermoFisher Titan Themis G2 60-300. In order to successfully implement the TEM characterizations, a carbon nanotube film (CNF, Nanjing XFNANO Materials Tech Co., Ltd.) was used as the current collector in cell assembly (denoted as Al|CNF/In cell). The Al|CNF/In cell was disassembled at a fully charged state in a glovebox, and the CNF electrode was ground into powder and mounted on the TEM holder.

### Electrochemical characterizations
The three-electrode beaker-type cells were fabricated with aluminium wires as counter and reference electrodes, molybdenum wire as the working electrode and chloroaluminate melt as the electrolyte in an argon-filled glovebox. An indium foil (Alfa Aesar, 99.99%) was immersed into the chloroaluminate melt to generate InCl. The beaker-type cells were placed into an oven and heated to different target temperatures from 110 to 190 °C. The CV curves were collected at a scan rate of 2 mV s$^{-1}$ using a Bio-logic SP-200 potentiostat. The home-made Swagelok® cell with a protection sheath was assembled using a commercial Al foil as the anode electrode, a glass fiber as the separator, the inorganic chloroaluminate melt as the electrolyte and molybdenum foil (or carbon cloth, CC; activated carbon cloth, ACC) as the positive current collector. The carbon fiber cloth was used as purchased (Shanghai He Sen Electric Co., Ltd.). The activated carbon cloth was obtained by treating the commercial carbon cloth at 450 °C for 2 h in the air. In the Al|Mo/In, Al|CC/In, and Al|ACC/In cells, the indium foil was placed between two glass fiber separators. In the Al|ACC/InCl cell, commercial InCl powder (Aladdin, 99.9%) was used as the cathode. In the Al|ACC/CuCl, Al|ACC/SnCl$_2$, Al|ACC/CrCl$_2$ and Al|ACC/MnCl$_2$ cells, commercial CuCl (Acros, 97%), SnCl$_2$ (Sigma-Aldrich, 98%), CrCl$_2$ (Behringer, 99.9%) and MnCl$_2$ (Sigma-Aldrich, 98%) powders were added to the cathode side. These cells were placed into a constant temperature testing chamber at 150 °C for electrochemical measurements. Different from the traditional method of cathode preparation (mixing the active materials, conductive carbon, and binder), we were able to use a very

simple method of simply spreading the commercial powders directly onto the surface of cathode current collector, because these materials can dissolve partially in the electrolyte. Once dissolved, they can diffuse to the conductive surface to participate in the reaction. This electrode configuration can greatly simplify the assembly of the cell. Galvanostatic charge-discharge measurements were performed using a multi-channel battery testing system (NEWARE). The charging rate capability of the cells was evaluated at varied charging rates from 0.5 to 20 mA cm$^{-2}$. In addition, because the indium foil spontaneously reacted with the chloroaluminate melt to supply InCl, the charge capacity cut-off of the Al|ACC/In cell was fixed to be 2 mAh cm$^{-2}$. At very high charging rates, the available areal capacity can be diffusion- or reaction-limited, for which the voltage limit was used as the cut-off. The cyclic voltammetry (CV) and electrochemical impedance spectra (EIS) were collected using a Bio-logic SP-200 potentiostat. Each EIS was measured in a frequency range from 20k Hz to 0.02 Hz with 91 points. The in-situ EIS measurement of the Al|ACC/In cell was conducted repetitively after a fixed capacity interval of 0.1 mAh cm$^{-2}$. The DRT analyses were performed by the MATLAB GUI toolbox developed by ref. [34]. For the freezing and thawing test, the Al|ACC/InCl cell was first tested at a discharging rate of 2 C and a charging rate of 4 C for 5 cycles. Then temperature of the cell dropped to room temperature by turning off the oven with the heat-off status maintained for 1 h. After that, the oven was turned on, and the temperature rose back to 150 °C.

## Data availability
The data that support the plots within this paper and other findings of this study are available from the corresponding author upon reasonable request. Source data are provided with this paper.

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

## Acknowledgements
This work was supported by the National Key R&D Program of China (grant no. 2022YFE0198600). We also thank the support from National Natural Science Foundation of China (NSFC) (grant no. 22075002, and 52103329); National Postdoctoral Program for Innovative Talents of China (grant no. BX2021002); and the China Postdoctoral Science Foundation (grant no. 2021M690194).

## Author contributions
Q.P. and J.M. conceived the concept. Q.P., J.M., and Y.Z. designed the experimental work. J.M., X.Y., and L.Z. prepared the electrolytes and performed the physical characterization of the electrolytes and electrodes. J.M., Z.X., and Y.J. performed electrochemical performance measurements. X.H. participated in electrochemical analysis. F.L. and H.S. participated in the ex-situ XRD and XPS studies, as well as electron microscopy characterizations. Q.P., J.M., and X.H. proposed the reaction mechanism. All authors have thoroughly discussed the analysis of the data. Q.P., J.M., X.Y., and Y.Z. wrote the manuscripts with contributions from all authors. Q.P. supervised the work.

## Competing interests
The authors declare no competing interests.
