## [Peer Review File · Nature Communications]

A solution-to-solid conversion chemistry enables ultrafast-charging and long-lived molten salt aluminium batteriesREVIEWER COMMENTS

Reviewer #1 (Remarks to the Author):

This manuscript reports a new rechargeable aluminum battery using the redox reaction between InCl and InCl₃ as the cathode reaction. The reported battery benefits from the distinct solubility of InCl and InCl₃ in the high-temperature molten salt electrolyte. The manuscript can be accepted for publication after addressing following comments:

- (1) The self-discharge property of the reported battery needs to be studied.
- (2) The rate capability tests only test high charging rates, not the discharging rates, what is the reason?
- (3) The experimental procedure stated the charging capacity in the cycling test was kept at 2 mAh/cm² (Figure 3d). But clearly the capacity of some high rate charging was below 2 mAh/cm². What is the cut-off for these charging processes?
- (4) In metal reacts with the molten salt to produce InCl, but which Al-containing species is the reactant, AlCl₄ or Al₂Cl₇?

Reviewer #2 (Remarks to the Author):

The manuscript presents an interesting electrochemical energy storage concept based on InCl/InCl₃ redox couple combined with Al plating/stripping process of chloroaluminate based molten salt. Considering the practical aspects of this concept in the context of achievable energy densities, I doubt that it can be of any use. The main problem is the low solubility of InCl in the chloroaluminate melt of about 3 wt%. This means that InCl should always be kept in a soluble state to support the claimed solution-to-solid conversion mechanism. Therefore, the chloroaluminate melt required to dissolve the InCl active material should be included in the calculation of the theoretical capacity of the active material, which in this case is InCl/AlCl₃-NaCl-KCl. My calculations show that the capacity of such an "active material" would be extremely small, about 10.7 mAh/g, resulting in a very small theoretical energy density of about 10.7 Wh/kg (considering the average discharge voltage of 1V and the fact that AlCl₃-NaCl-KCl is a kind of active material, used for Al plating).

On the other hand, I tend to believe that the InCl/InCl₃ solid-solid conversion reaction could be quite reversible with low polarization at such high temperatures. In this case, one should not use large amounts of the chloroaluminate melts needed to dissolve InCl. Instead, a very small amount of salt melt can be used (the amount needed to match the capacity at the InCl/InCl₃ electrode), maximizing the energy density of the system. According to my calculations, the theoretical energy density in this case is about 200 Wh/kg.

In conclusion, the "solution to solid" does not make much sense from an energy density point of view. The authors could try to prove the solid-to-solid mechanism by minimizing the amount of chloroaluminate melt needed to dissolve InCl. However, the paper should be rewritten accordingly. Another important point is the demonstration that there is no electrochemical plating of In⁺ on the Al foil during charging. Therefore, the authors must characterize the Al foil as well after charging.

Reviewer #3 (Remarks to the Author):

The manuscript by Meng et al. presents a study on the understanding of solution-to-solid conversion chemistry in aluminum-ion batteries which employ manipulated solubility as cathodes and molten salt electrolytes. This concept is interesting and creative.

However, this is the first time that I had doubts on creditability of the authors. The manuscript has many false claims in the introduction, as well as a clear lack of a rigorous presentation in C-rate, capacity, and other device data. It made me second guess whether this work can be trusted or whether it is creditable. If everything is true, I believe that i) the generality of the proposed concept is not yet sufficiently shown and that ii) the battery performance data of this work is limited.

I propose several remarks below to help the authors re-do their writing. As far as creditability, if they decide to revise and submit again, I would like to see a written statement from the corresponding author that he has double-checked all the work/all raw data from his student(s).

1) Literature discussion is FAR FROM adequate. In the second paragraph on page 2, some statements, such as "the development of viable RABs faces several issues including limited charge storage capacity, poor reaction kinetics and very short cycling life" and "the low capacity of 60~100 mAh g⁻¹ and significant electrode expansion featured by the anion-insertion chemistry discourages their application in large-format cells", are false. In fact, several tens of thousands of cycles (250,000 cycles, Chen H. et al. Science Advances. 2017. 12. eaao7233) and large storage capacity (200 mAh/g, Shen X. et al. Nature Communications. 2021.1.820) in Al-ion batteries have already been proven. The electrode expansion is not a problem in ref 14. It has been solved by using a graphene cathode in ref 15. Additionally, Fig. 5e should be reorganized.

2) Why are all discharging rate less than the charging rate? What is the constant charging-discharging and different discharging rate behaviors of the battery (storage capacity, voltage profiles, rate performance and cycle life)? What is the discharging rate in Fig.3f? How does the discharging rate affect cycle performance?

3) There is a lack of discussion in the last section (cells directly using InCl cathode). Authors mentioned that "In order to realize this solution-to-solid conversion chemistry on the basis of conventional battery production infrastructure, cells were fabricated using powder InCl cathodes directly". However, this part dose not discuss the comparison of the two battery systems, but lists some data. In addition, mass is used to calculate the current density in this part, and area is used to calculate in the previous part. The full manuscript is a mess in capacity; it has no whatsoever explanations.

4) Fig.3b, one scan takes roughly 8000 sec. Why 0.2 mV/s as the scan rate? what is the rate influence to the redox reactions?

5) In Fig.4a, why is the discharge time twice as long as the charging time (100 min vs. 50 min)?

6) How are the powders (InCl, CuCl, SnCl₂, CrCl₂ and MnCl₂) prepared into the cathodes?

7) Please describe the calculation method of specific surface area and the C-rate in this case.

8) How to explain the low coulombic efficiency in Supplementary Fig.4d.

Reviewer #1 (Remarks to the Author):

This manuscript reports a new rechargeable aluminum battery using the redox reaction between InCl and InCl₃ as the cathode reaction. The reported battery benefits from the distinct solubility of InCl and InCl₃ in the high-temperature molten salt electrolyte. The manuscript can be accepted for publication after addressing following comments:

Our response: We fully appreciate the reviewer's thoughtful and encouraging comments about our manuscript, and offering the opportunity to address and clarify the issues raised in the report. Our responses to the points raised in the report are described below following the reviewer's comments.

(1) The self-discharge property of the reported battery needs to be studied.

Our response:

We appreciate the reviewer's valuable suggestion on studying the self-discharging behavior. This is indeed important for a battery with new chemistry and we have overlooked this measurement. To answer the question, first, we have now investigated the self-discharge property of the Al|ACC/InCl cell operated at 150 °C. As shown in Figure R1, the cell was cycled in a charging rate of 1C and a discharging rate of 0.5C. The discharge capacity in the second cycle is ~314 mAh g⁻¹. In the third cycle, the cell was charged to 1.3 V at a rate of 1C, kept at 150 °C for 4 h, and then discharged to 0.4 V at a rate of 0.5C. The cell can sustain a discharge capacity of 299 mAh g⁻¹, displaying a high capacity retention of ~95.2% compared with the discharge capacity in the second cycle. This indicates that our cell exhibits a low self-discharge rate at the operation temperature, attributed to the low solubility of solid InCl₃ in the molten salt electrolyte. Second, we further note that the molten salt batteries have a universal advantage that they can be scheduled to cool down to temperatures below or around the melting point of the electrolytes when not in use (on the shelf); in this way, the batteries can maintain close-to-none self-discharge due to the elimination of any material cross-over between the two electrodes.

We have added this result and discussions in our revised manuscript.

Figure R1. The self-discharge test of the Al|ACC/InCl cell operating at 150 °C. (a) The voltage profile of the Al|ACC/InCl cell during the charge and discharge process with an interval after charging. A charging rate of 1C and a discharging rate of 0.5C are performed, consistent with other cycling performance tests of the Al|ACC/InCl cell. (b) The corresponding voltage profiles at the second and third cycles.

(2) The rate capability tests only test high charging rates, not the discharging rates, what is the reason?

Our response:

We thank the reviewer for raising concerns on the discharging rate capacity of the cell. As we indicated in the title, abstract and body of the manuscript, we stressed that the new battery system has fast-charging capability. The underlying reason why we emphasized only on the charging rate capacity is that the redox reaction in the cell is asymmetric, meaning the reaction pathways and their kinetics on charge and discharge are different. To demonstrate this, in the manuscript, we first showed our investigation of the nature of the indium electrochemistry using a three-electrode beaker-type cell with the chloroaluminate melt electrolyte. As shown in Fig. 2c, the cyclic voltammetry measurement shows sharp anodic peaks but relatively broad cathodic peaks at a range of operation temperatures, indicating an asymmetric reaction between the highly soluble InCl (In⁺) and sparingly soluble InCl₃ (In³⁺). Featured with this solution-to-solid conversion chemistry, our cell exhibits remarkable charging rate capability. Therefore, in our manuscript, we mainly focus on the charging rate capability.

Nevertheless, as inspired by the reviewer, it would be interesting to test the limit of the discharging capability. Therefore, we have now tested the discharging rate performance of our Al|ACC/In cell with symmetric current densities ranging from 0.5 to 10 mA cm⁻² at 150 °C (Figure R2). It is clear that the cell retains rather high areal

discharge capacities of 1.93, 1.66 and 1.42 mAh cm⁻² with low voltage polarization at low current densities of 1, 2 and 3 mA cm⁻², respectively. At higher current densities of 5 and 10 mA cm⁻², the cell shows decent capacities of 1.12 and 0.84 mAh cm⁻², although the voltage polarization is higher. Therefore, we conclude that the cell shows an inferior discharging rate performance than the charging, but nevertheless satisfactory for practical applications particularly compared with the conversion reaction-based aluminium batteries using ionic liquid electrolytes. This is fundamentally owed to the use of chloroaluminate melt electrolyte that shows very high desolvation kinetics, and to the presence of dissolved InCl as the discharge product. Note that it is also clear that the overpotential on charge is higher than that on discharge at the same rates, again confirming asymmetric reactions.

We have added this data and the discussions in the revised manuscript.

Figure R2. (a) Rate performance of the Al|ACC/In cell at different rates ranging from 0.5 to 10 mA cm⁻², and (b) the corresponding voltage profiles at different rates. The cell uses the same rates for the discharge and charge at all cycle numbers.

(3) The experimental procedure stated the charging capacity in the cycling test was kept at 2 mAh/cm² (Figure 3d). But clearly the capacity of some high rate charging was below 2 mAh/cm². What is the cut-off for these charging processes?

Our response:

We appreciate the reviewer raised the concern on the procedure of testing the charging rate capability. Indeed, we admit that it was confusing to use “limited charge capacity of 2 mAh/cm²” in the manuscript, and using “capacity cut-off” is more appropriate to describe our charging protocol. In fact, we use a protocol of *either* capacity cut-off of 2 mAh/cm² *or* a voltage cut-off (now detailed in the Method section).

To further clarify why this protocol was used, we would like to draw attention to the fact the Al|ACC/In cell (shown in Figure 3) has excess amount of dissolved InCl in the electrolyte as the dissolution of In⁺ (by reacting an indium foil with the Al_xCl_y⁻ anions) is a dynamic process which continues as its oxidation occurs. In another word, when the In⁺ is consumed during charging, the spontaneous reaction between indium metal and chloroaluminate anions occurs to supply more InCl into the electrolyte. Therefore, at a low charging rate, the formation of solid InCl₃ product, that is, its electrodeposition on the carbon surface, is limited not by the amount of InCl available, but by the available surface area of the carbon. Therefore, the reaction can continue for long period and the areal capacity can be well over 2 mAh cm⁻², given the high surface area of the electrode we used. However, at very high charging rates, the available areal capacity can be diffusion- or reaction-limited, and is thus lower than 2 mAh cm⁻², for which the voltage limit become effectively the cut-off. This is indeed what we observed from the charging rate measurement (Figure 3d). We therefore set a capacity cut-off of 2 mAh cm⁻² to avoid this and to make fair comparison with the performance at very high rates.

We do note that the unlimited charging capacity at low rates does not mean shuttling of active materials (as in the case for Li-S batteries), because the discharge capacity would've been also unlimited and close to that of the previous charging capacity. This is similar to a metal-air battery measurement where the areal discharging capacity is limited for simplified measurement of the voltage profiles.

We feel that it was indeed confusing due to the special configuration of the Al|ACC/In cell, and therefore, we have now revised the description and added the discussions in the revised manuscript.

(4) In metal reacts with the molten salt to produce InCl, but which Al-containing species is the reactant, AlCl₄ or Al₂Cl₇?

Our response:

We thank the reviewer for bringing up the question on the reaction mechanism. We apologize for not being clear on which species is exactly reacting with the indium. We understand that the chloroaluminate melt contains AlCl₄⁻ and Al₂Cl₇⁻ and other long-chain Al₃Cl₁₀⁻ species (Nature, 2022, 608, 704-711.). The reaction with indium metal requires breaking of Al-Cl bonds and therefore, we propose that the species that has weak Al-Cl bonding would preferentially react with the indium metal. For such, we

learned that the longer-chain $Al_3Cl_{10}^-$ or $Al_2Cl_7^-$ species has weaker Al-Cl bridging bonds than the $AlCl_4^-$ species (Nature, 2022, 608, 704-711.). Therefore, we believe it is the $Al_3Cl_{10}^-$ or $Al_2Cl_7^-$ species that preferentially react with indium. To reflect this, we have now revised the reaction equation between indium metal and the chloroaluminate salt in our manuscript. The equation is as follows:

We have now added discussions on this point in the revised manuscript.

Reviewer #2 (Remarks to the Author):

The manuscript presents an interesting electrochemical energy storage concept based on InCl/InCl₃ redox couple combined with Al plating/stripping process of chloroaluminate based molten salt. Considering the practical aspects of this concept in the context of achievable energy densities, I doubt that it can be of any use. The main problem is the low solubility of InCl in the chloroaluminate melt of about 3 wt%. This means that InCl should always be kept in a soluble state to support the claimed solution-to-solid conversion mechanism. Therefore, the chloroaluminate melt required to dissolve the InCl active material should be included in the calculation of the theoretical capacity of the active material, which in this case is InCl/AlCl₃-NaCl-KCl. My calculations show that the capacity of such an "active material" would be extremely small, about 10.7 mAh/g, resulting in a very small theoretical energy density of about 10.7 Wh/kg (considering the average discharge voltage of 1V and the fact that AlCl₃-NaCl-KCl is a kind of active material, used for Al plating).

On the other hand, I tend to believe that the InCl/InCl₃ solid-solid conversion reaction could be quite reversible with low polarization at such high temperatures. In this case, one should not use large amounts of the chloroaluminate melts needed to dissolve InCl. Instead, a very small amount of salt melt can be used (the amount needed to match the capacity at the InCl/InCl₃ electrode), maximizing the energy density of the system. According to my calculations, the theoretical energy density in this case is about 200 Wh/kg. In conclusion, the "solution to solid" does not make much sense from an energy density point of view. The authors could try to prove the solid-to-solid mechanism by minimizing the amount of chloroaluminate melt needed to dissolve InCl. However, the paper should be rewritten accordingly.

Our response:

We appreciate the reviewer's comment that this is an interesting electrochemical energy storage concept.

We would like to reiterate that the major novelty of this work is a successful demonstration of solution-to-solid conversion chemistry for rechargeable aluminium batteries featuring ultrafast charging and long-lived performance. Unlike the conventional solid-to-solid conversion reactions, this solution-to-solid reaction features intrinsic fast kinetics and structural self-healing characteristics. To our best knowledge, it is the first time to discover and propose this unique reaction mechanism in RABs. As

a proof-of-concept, we disclose a highly reversible redox couple – the highly soluble InCl and the sparingly soluble InCl₃. We have demonstrated that the Al|ACC/InCl cell exhibits a high areal capacity, low cell overpotential, good charging rate capability and cycling stability.

Further, we appreciate the reviewer's comment on the practical energy density of the cell as this inspired us to further examine the potential of the battery in term of the energy density. However, we respectfully disagree with the reviewer on the practical energy density of the cell, given the following facts and arguments:

First, we fully agree with the reviewer that if all the In(I) in the cell has to be fully dissolved to undergo the solution-to-solid reaction, we have to include the mass of the electrolyte for calculating the overall energy density; and we appreciate that a solid-to-solid reaction would leads to much higher energy density. However, we argue that in a practical operation, given a fixed area of the conducting surface in the electrode, not all the In(I) in the cell would participate in the oxidation at once, but in fact, the In(I) reacts at a fixed rate on charge, which means that the dissolved In(I) can be dynamically formed as the reaction continues for the solution-to-solid reaction scheme to be true. In another word, after the In(I) in the electric double layer close to the carbon surface has completed its reaction, its concentration would decrease, which dynamically drives further dissolution of In(I) from the solid InCl present in the electrode. With this scheme, the solid InCl would eventually find its way to participate in the solution-to-solid reaction. Therefore, in principle not all In(I) has to be pre-dissolved in the electrolyte for the solution-to-solid reaction to predominantly occur. In fact, this theory finds its application in “semi-solid” redox flow batteries (that rely on liquid phase reaction, but use solid particles that have certain solubility in the electrolyte), and lithium-sulfur batteries. For example, in lithium-sulfur batteries, the sulfur undergoes dissolution and precipitation reaction on discharge, and in fact not all polysulfides are dissolved and reacted at once.

Second, with that said, the solution-to-solid reaction scheme is fundamentally true for the system even if not all InCl is dissolved to begin with. Nevertheless, as respectfully inspired by the reviewer's constructive comments, we also understand that the correctness of the statement essentially comes down to the competition of the In(I) dissolution and its oxidation reaction. In principle, if the In(I) dissolves at a faster rate than its oxidation reaction, it would undergo majorly a solution-to-solid reaction;

otherwise, there would be a mixed reaction scheme, that is, solid-to-solution and solution-to-solid reactions occurring in tandem or in parallel.

→ To confirm the statement and examine the competition of the dissolution and reaction in a practical cell, we have now carried out the rate performance of the Al|ACC/InCl cells with high InCl mass loadings and less electrolyte. In the experiment, the mass of molten salt electrolyte added in each cell was fixed to be 100 mg, and the mass of InCl was varied. We first examined the performances using a high InCl loading of 10 mg. At the operation temperature of 150 °C, an InCl mass of 10 mg does not fully dissolve into the electrolyte (based on the measured solubility shown in Fig. 2d). To be exact, about 60% of the active InCl material exists in solid state. As shown in Figure R3a,b, the Al|ACC/InCl cell shows a high discharge capacity of $\sim 261 \text{ mAh g}^{-1}$ (compared to 327 mAh g^{-1} capacity at a InCl loading of $\sim 2 \text{ mg cm}^{-2}$ shown in Figure 5c), a high Coulombic efficiency of $\sim 96\%$ and small overpotential of $\sim 25 \text{ mV}$ at a rate of 0.2C. The high InCl utilization rate obtained shows that even if 60% of the InCl is present as solid, high InCl utilization of 73% can be achieved. When increasing the current density to 1C, the cell exhibits a reversible capacity of $\sim 100 \text{ mAh g}^{-1}$ and good cycling stability. Given that such high loading can experience capacity limitation from the mass diffusion problem, it is thus not possible to isolate and determine the limits from the reaction kinetics; nevertheless, the utilization rate of 73% at 0.2C can be considered to be high. This experiment proves that not all In(I) has to be dissolved to participate in the solution-to-solid reaction, and it is possible to have solid-to-solution and solution-to-solid reactions occurring in tandem or in parallel at high charging rates.

→ We also further show that even at a higher InCl mass loading of 20 mg, wherein $\sim 80\%$ of active InCl material exists in solid state, the Al|ACC/InCl cell shows a high specific capacity of $\sim 200 \text{ mAh g}^{-1}$ at 0.2 C. At a high charging rate of 1C, the cell can sustain a capacity of 75 mAh g^{-1} . We envision that in such case, we are observing a higher percentage of solid-to-solid reaction which is relatively slow and causes higher voltage polarization (Figure R3c,d).

Taken together, these results indicate that our Al|ACC/InCl cells with high InCl mass loadings and low fraction of dissolved InCl can still work well by a mechanism of solid-to-solution and solution-to-solid conversion reaction in tandem or in parallel. The practical energy density is surely associated with the rate applied to the cell. The conclusion of the rapid solution-to-solid reaction is fundamentally true but apparently needs modification by incorporating the thoughts gathered with the new data. In terms

of calculating the practical cell-level energy density, it is thus not plausible to include the total mass of the electrolyte required for full dissolution. Therefore, we respectfully disagree with the reviewer's comment on the lack of significance of our solution-to-solid reaction concept, but we have also now incorporated the thoughts on solid-to-solid reaction inspired by the reviewer and greatly modified the reaction scheme in the revised manuscript.

Figure R3. The rate performance of the Al|ACC/InCl cells with high InCl mass loadings. (a) The rate performance of the Al|ACC/InCl cell using an active InCl mass of 10 mg and an electrolyte mass of 100 mg, with varied rates from 0.2C to 1C at 150 °C, and (b) the corresponding voltage profiles at different rates. (c) The rate performance of the Al|ACC/InCl cell using an active InCl mass of 20 mg and an electrolyte mass of 100 mg at 150 °C, with varied rates from 0.2C to 1C, and (d) the corresponding voltage profiles at different rates. The cell uses same rates for the discharge and charge at all cycle numbers.

Third, given the above arguments, nevertheless, we further propose that it is important to further increase the solubility of the active species in the chloroaluminate melt to increase the utilization at high charging rates thus improving the overall energy density. There can be two efficient methods to increase the solubility of InCl in the chloroaluminate melt. (1) Increasing the operation temperature can greatly improve the solubility of InCl. As shown in Fig. 2d, the indium solubility can increase up to 5.05 wt% as the operation temperature increases to 190 °C. The reaction kinetics of indium electrochemistry can also be greatly enhanced as demonstrated in Fig. 2c. This definitely points to a rational direction for practical applications. (2) Increasing the Lewis acidity of the chloroaluminate melt is beneficial for increasing the In(+)

solubility. As indicated by the equation of $\text{Al}_n\text{Cl}_{3n+1}^- + 3(n-1)\text{In}(\text{s}) \rightarrow 3(n-1)\text{InCl}(\text{sol}) + (n-1)\text{Al}(\text{s}) + \text{AlCl}_4^-$ ($n \geq 2$), an increased ratio of the higher-order $\text{Al}_n\text{Cl}_{3n+1}^-$ species (that is, increased Lewis acidity) facilitates the generation of InCl . We have now added the discussions on this point in the revised manuscript.

Another important point is the demonstration that there is no electrochemical plating of In^+ on the Al foil during charging. Therefore, the authors must characterize the Al foil as well after charging.

Our response:

We agree with the reviewer that it is necessary to prove that the deposits are pure aluminium and free of indium. We have now performed the measurement by examining an Al|Mo/In cell after cycling at 150 °C. After deposition at a current density of 1 mA cm⁻² for 5 mAh cm⁻², we can observe that the Al deposits show a morphology of compact crystals with well-defined facets. The elemental analysis with EDX spectra and mapping show that there is none indium signal on the Al deposits to the resolution and detection limit of our EDX detector (Figure R4). This clearly shows that indium is not co-deposited with aluminium. In fact, from the point of thermodynamic redox potential in theory, we showed that indium can spontaneously reduce the chloroaluminate species to generate soluble monovalent In^+ and Al, and this shows a more negative $\text{In}(\text{I})/\text{In}(\text{0})$ couple than the $\text{Al}(\text{III})/\text{Al}(\text{0})$ couple. Therefore, the preferential formation of Al deposit is a thermodynamically driven process. We have now added the data and discussed this thoroughly in our revised manuscript.

Figure R4. The SEM images (a-c) and the corresponding EDX mapping (Al, In) (d,e), EDX spectrum (f) of the electroplated Al on Mo in the NaCl-KCl-AlCl₃ electrolyte at 150 °C with the current of 1 mA cm⁻² for an areal capacity of 5 mAh cm⁻².

Reviewer #3 (Remarks to the Author):

The manuscript by Meng et al. presents a study on the understanding of solution-to-solid conversion chemistry in aluminum-ion batteries which employ manipulated solubility as cathodes and molten salt electrolytes. This concept is interesting and creative.

Our response:

We very much appreciate the reviewer's general comment that the concept of solution-to-solid reaction is interesting and creative, using the cathodes with manipulated solubility and the molten salt electrolyte.

However, this is the first time that I had doubts on creditability of the authors. The manuscript has many false claims in the introduction, as well as a clear lack of a rigorous presentation in C-rate, capacity, and other device data. It made me second guess whether this work can be trusted or whether it is creditable. If everything is true, I believe that i) the generality of the proposed concept is not yet sufficiently shown and that ii) the battery performance data of this work is limited. I propose several remarks below to help the authors re-do their writing. As far as creditability, if they decide to revise and submit again, I would like to see a written statement from the corresponding author that he has double-checked all the work/all raw data from his student(s).

Our response:

First, we apologize for expressing some inaccurate claims in the introduction and not being crystal clear on presentations of our data. We have now added the key literatures mentioned by the reviewer into the introduction part and updated the related discussions.

Second, we further feel sorry to learn that the presentation of C-rate, capacity is confusing and perhaps misleading as the reviewer suggested. However, we would like to draw attention to the fact that the demonstrated chemistry is a rather unique solution-to-solid reaction which involves a solution phase as the product of one end and a solid phase as the product of another. This chemistry is unlike a standard intercalation chemistry like LiFePO_4 , nor a standard conversion chemistry like lithium-sulfur battery, which have reached universal standards with consensus on presenting the C-rate, specific capacity, areal capacity, rate performances etc. In these batteries, we simply use mass-based specific capacity mAh/g for understanding on the material level, and use areal capacity mAh/cm^2 for understanding on the electrode level.

→ Rather, we can consider the current solution-to-solid chemistry on par with a gas-battery, for example a Li-air battery that bases on the conversion between oxygen and $\text{Li}_2\text{O}/\text{Li}_2\text{O}_2$. For such chemistry, in the early days of research, one may present the capacity based on the mass of oxygen (which would be not meaningful as oxygen can be in excess), or on the mass of the carbon substrate materials (for the reason that the discharge capacity is limited by the available surface area), or simply on the geometric area of the electrode (to be universal across studies, but regardless of the underlying materials). Therefore, it is hard to use universal standards for presentation. In our case, in the first part of the manuscript where we used an indium metal in the cell to dynamically supply dissolved In(I) , the In(I) can be considered in excess and the available carbon surface is limited; therefore, we used the “areal” matrices to show the fundamental electrochemical behavior of the $\text{In(I)}/\text{In(III)}$ redox couple. In the second part of the manuscript, we directly used InCl as the cathode to fit in the evaluation framework of a standard battery, so that the mass of InCl is limited while the carbon surface is potentially in excess; therefore, we used the mass-based specific capacity, rates etc.

→ Nevertheless, we are not stating this to justify that we could overlook our unclear presentation of some of the data in the manuscript. On basis of the reviewer’s suggestion, we have discussed these considerations above and reinterpreted the C-rate, capacity, and other electrochemical tests for our cells in order to clearly convey to the readers. Our responses to the specific points raised by reviewer are described below in detail. We believe that these revisions and improvements will make the revised manuscript more reasonable. We appreciate the reviewer’s constructive comment again.

Third, in the meanwhile, we are surprised to receive the comment on the credibility of the co-authors, and the work itself. We understand that there is disagreement on presentation of the data as articulated above; however, the data we collected and showed in the manuscript were not treated with carelessness. On basis of the reviewer’s suggestion, we have now double-checked all raw data shown in our manuscript. All the showed results and the added data during the revision are correct. Nevertheless, to fully show the creditability of our work, we have now uploaded our raw data as supporting files to the system.

Fourth, in terms of the generality of the proposed concept, we do agree that it is sufficiently shown in the manuscript. However, we did not aim to extensively show

evidence of identifying materials that function excellent based on the solution-to-solid chemistry, but rather, by extending to other redox couples, we aimed to identify and articulate the basic criteria for the redox couples to function via a solution-to-solid mechanism in a full RAB. The question has to do with the redox potential of the pair so that it generates meaningful voltage and is stable with the Al anode, and with the solubility of the pair so that it favors high reaction kinetics. To this end, the CrCl_2 is the only material that functions out of the four studied materials CuCl , SnCl_2 , CrCl_2 and MnCl_2 . For extensive study on the CrCl_2 , we believe it is beyond the scope of the current manuscript, and will be discussed in a separate report. We have now revised our language on this part to reflect what the reviewer suggested.

Last, in terms of the comment on the battery performance is limited, we kindly guess the reviewer may mean the long-term cycling stability and the performance at high loading of materials as discussed below:

→ With respect to the cycling stability, our $\text{Al}|\text{ACC}/\text{InCl}$ cell exhibits a small cell overpotential of only 35 mV at 1C rate and 150 °C. The cells show excellent charging rate capability and remarkable cycling stability with almost no capacity fade over 500 cycles at a 20C charging rate (as shown in Fig. 5). Admittedly, the demonstrated cycling life is shorter than the AlCl_4^- intercalation based graphite cathodes as the reviewer mentioned below, it is superior than most conventional conversion cathodes in previous reports. We have now added the discussion and comparison with previous reports on graphites in the revised manuscript.

→ With respect to the high loading performances, indeed this is important for a conversion type chemistry and we did not study extensively in this aspect. Therefore, we have now carried out the rate performance of the $\text{Al}|\text{ACC}/\text{InCl}$ cells with high InCl mass loadings and less electrolyte. We examined the performances using a high InCl loading of 10 mg and 20 mg with a fixed amount of electrolyte at 150 °C. As shown in the figure below (Figure R3a,b) the $\text{Al}|\text{ACC}/\text{InCl}$ cell shows a high discharge capacity of $\sim 261 \text{ mAh g}^{-1}$, a high Coulombic efficiency of $\sim 96\%$ and small overpotential of $\sim 20 \text{ mV}$ at a rate of 0.2C. When increasing the current density to 1C, the cell exhibits a reversible capacity of $\sim 100 \text{ mAh g}^{-1}$ and good cycling stability. We also further show that even at a higher InCl mass loading of 20 mg, the $\text{Al}|\text{ACC}/\text{InCl}$ cell shows a high specific capacity of $\sim 200 \text{ mAh g}^{-1}$ at 0.2 C. At a high charging rate of 1C, the cell can sustain a capacity of 75 mAh g^{-1} . Same as the response to Reviewer 2, we have now

added this discussion in the revised manuscript.

Figure R3. The rate performance of the Al|ACC/InCl cells with high InCl mass loadings. (a) The rate performance of the Al|ACC/InCl cell using an active InCl mass of 10 mg and an electrolyte mass of 100 mg, with varied rates from 0.2C to 1C at 150 °C, and (b) the corresponding voltage profiles at different rates. (c) The rate performance of the Al|ACC/InCl cell using an active InCl mass of 20 mg and an electrolyte mass of 100 mg at 150 °C, with varied rates from 0.2C to 1C, and (d) the corresponding voltage profiles at different rates. The cell uses same rates for the discharge and charge at all cycle numbers.

1) Literature discussion is FAR FROM adequate. In the second paragraph on page 2, some statements, such as “the development of viable RABs faces several issues including limited charge storage capacity, poor reaction kinetics and very short cycling life” and “the low capacity of 60~100 mAh g⁻¹ and significant electrode expansion featured by the anion-insertion chemistry discourages their application in large-format cells”, are false. In fact, several tens of thousands of cycles (250,000 cycles, Chen H. et al. Science Advances. 2017. 12. eaao7233) and large storage capacity (200 mAh/g, Shen X. et al. Nature Communications. 2021.1.820) in Al-ion batteries have already been proven. The electrode expansion is not a problem in ref 14. It has been solved by using a graphene cathode in ref 15. Additionally, Fig. 5e should be reorganized.

Our response:

We appreciate the reviewer’s constructive comment. We agree and apologize that the efforts in improving the cycling life and solving the volume expansion of graphite

electrodes have been largely overlooked in the introduction.

We have rewritten this part by incorporating these efforts. In addition, we have added the two key references about rationally designed graphene cathodes in our revised manuscript. The Fig. 5e is now updated. **We have now discussed this thoroughly in the manuscript.**

2) Why are all discharging rate less than the charging rate? What is the constant charging-discharging and different discharging rate behaviors of the battery (storage capacity, voltage profiles, rate performance and cycle life)? What is the discharging rate in Fig.3f? How does the discharging rate affect cycle performance?

Our response:

We appreciate the reviewer's comment. We did not intentionally use lower discharging rates than the charging rates in the performance measurements, but in fact we used the charging or discharging rates individually based on the nature of its charging rate capability and its discharging rate capability. In another word, the cell with a solution-to-solid reaction mechanism exhibits asymmetric charging/discharging rate capabilities. This is fundamentally ascribed to the intrinsic asymmetric reaction path and kinetics on discharge and charge. Indeed, the nature of the indium electrochemistry in chloroaluminate melt electrolyte was first investigated using a three-electrode beaker-type cell. As shown in Fig. 2c, the cyclic voltammetry measurement shows sharp anodic peaks and relatively broad cathodic peaks. This asymmetric phenomenon is attributed to the asymmetric redox process between highly soluble InCl (In^+) and sparingly soluble InCl_3 (In^{3+}). That is, the charging reaction kinetics from solution to solid is higher than the discharging reaction kinetics from solid to solution. Featured with this solution-to-solid conversion chemistry, the cell exhibits remarkable charging rate capability. Therefore, in our manuscript, we mainly focus on the charging rate capability of our cells.

Further, inspired by the reviewer's suggestion, we also tested the constant charging-discharging rate performances of our Al|ACC/In cell ranging from 0.5 to 10 mA cm^{-2} at 150 °C (Figure R2). It is clear that the cell retains rather high areal discharge capacities of 1.93, 1.66 and 1.42 mAh cm^{-2} with low voltage polarization at a current density of 1, 2 and 3 mA/cm^2 . At higher current densities of 5 and 10 mA cm^{-2} , the cell shows decent capacities of 1.12 and 0.84 mAh cm^{-2} , although the voltage polarization

is higher. Therefore, we conclude that the cell shows an inferior discharging rate performance than the charging, but nevertheless satisfactory for practical applications particularly compared with the conversion reaction-based aluminium batteries using ionic liquid electrolytes. This is fundamentally owed to the use of chloroaluminate melt electrolyte that shows very high desolvation kinetics, and to the presence of dissolved InCl as the discharge product. Note that it is also clear that the overpotential on charge is higher than that on discharge at the same rates, again confirming asymmetric reactions. We have added this result and corresponding discussions into our revised manuscript.

We apologize for not making it clear on the discharging rate in Fig. 3f. The discharging rate is 0.5 mA cm^{-2} , consistent with other cycling performance tests of the Al|ACC/In cell. We have added this information in the caption of Fig. 3f.

Figure R2. (a) Rate performance of the Al|ACC/In cell at different rates ranging from 0.5 to 10 mA cm^{-2} , and (b) the corresponding voltage profiles at different rates. The cell uses the same rates for the discharge and charge at all cycle numbers.

3) There is a lack of discussion in the last section (cells directly using InCl cathode). Authors mentioned that “In order to realize this solution-to-solid conversion chemistry on the basis of conventional battery production infrastructure, cells were fabricated using powder InCl cathodes directly”. However, this part does not discuss the comparison of the two battery systems, but lists some data. In addition, mass is used to calculate the current density in this part, and area is used to calculate in the previous part. The full manuscript is a mess in capacity; it has no whatsoever explanations.

Our response:

We appreciate the reviewer's constructive comments. We apologize for not being clear in why we have investigated two battery systems and the difference in describing the performance matrices. The comparison of the two battery systems are listed herein for the information of the reviewer. In the first part of the manuscript, we used an indium metal in the cell to dynamically supply dissolved In(I), and the In(I) can be considered in excess and the available carbon surface is limited; therefore, we used the "areal" matrices to show the fundamental electrochemical behavior of the In(I)/In(III) redox couple. In the second part of the manuscript, we directly used InCl as the cathode to fit in the evaluation framework of a standard battery, so that the mass of InCl is limited while the carbon surface is potentially in excess; therefore, we used the mass-based specific capacity and rates etc. In brief, this difference of the two cells on current density and capacity depends on which one is the limiting factor for capacity. **We have now added this discussion in our revised manuscript.**

4) Fig.3b, one scan takes roughly 8000 sec. Why 0.2 mV/s as the scan rate? what is the rate influence to the redox reactions?

Our response:

We appreciate the reviewer's comment. We understand that a faster CV scan rate leads to decrease in the size of the diffusion layer at the proximity of the electrode surface. In our experiment, we used a low scan rate of 0.2 mV s⁻¹ to investigate the nature of the reduction and oxidation reactions at the condition of adequate diffusion.

To answer the question of the rate influence to the redox reactions, we have now performed CV measurements at varied scan rates. As shown in Figure R5, high scan rates over 1 mV s⁻¹ lead to insufficient reaction of the active material and obvious cell voltage polarization. In our Al|ACC/In cell, the In metal can supply the InCl into the electrolyte when the active InCl was oxidized to solid InCl₃ during charging. At high scan rates, the size of the diffusion layer becomes small, and the electrochemical reaction process is diffusion-controlled. **We have now discussed this in our revised manuscript.**

Figure R5. CV curves of the Al|ACC/In cell at different scan rates ranging from 1 to 10 mV s⁻¹.

5) In Fig.4a, why is the discharge time twice as long as the charging time (100 min vs. 50 min)?

Our response:

We apologize for not stating this clearly the rates used for the *ex-situ* XRD characterizations. In fact, we used a discharging rate that is half that for charge, to ensure full reaction on discharge in our customized XRD cell. Again, as described above in question 2, this is attributed to the asymmetric reaction mechanisms during reaction. **We have now clearly stated this in the revised manuscript.**

6) How are the powders (InCl, CuCl, SnCl₂, CrCl₂ and MnCl₂) prepared into the cathodes?

Our response:

We appreciate the reviewer's comment. Different from the traditional method of cathode preparation (mixing the active materials, conductive carbon and binder), we were able to use a very simple method of simply spreading the commercial powders directly onto the surface of cathode current collector, because these materials can dissolve partially in the electrolyte. Once dissolved, they can diffuse to the conductive surface to participate in the reaction. This electrode configuration can greatly simplify the assembly of the cell. **We have now added the detailed information in our revised manuscript.**

7) Please describe the calculation method of specific surface area and the C-rate in this case.

Our response:

We appreciate the reviewer's suggestion. Specific surface area was measured by the isothermal nitrogen adsorption and calculated based on the Brunauer-Emmett-Teller (BET) method.

The Al|ACC/InCl cells were fabricated by directly using powder InCl cathodes. The theoretical gravimetric capacity of InCl is calculated to be 357 mAh g⁻¹. Therefore, 1C means a current density of 357 mA g⁻¹. We have now added this information in our revised manuscript.

8) How to explain the low coulombic efficiency in Supplementary Fig.4d.

Our response:

We appreciate the reviewer's question. The original Supplementary Figure 4d (now Supplementary Figure 5f) shows the voltage profiles for a Al|ACC/In cell using organic ionic liquid electrolyte at 150 °C. The low coulombic efficiency is attributed to a rather high solubility of the generated InCl₃ in hot ionic liquids, leading to serious shuttle effect during charging. We have now discussed this in our revised manuscript.

REVIEWERS' COMMENTS

Reviewer #1 (Remarks to the Author):

I believe the authors fully addressed the comments from the reviewers. The revised manuscript is much better presented with clarity.

Reviewer #3 (Remarks to the Author):

I have read the responses by authors and am generally happy about the revision.

I do notice a new thing in Figure R5. It seems there are more than one redox reaction, different from Figure 2C or 3B, please explain what they are.

Once this new question is taken care of, I am recommending the manuscript to be accepted.

Reviewer #1 (Remarks to the Author):

I believe the authors fully addressed the comments from the reviewers. The revised manuscript is much better presented with clarity.

Our response: We thank the reviewer again for their thoughtful comments about our manuscript.

Reviewer #3 (Remarks to the Author):

I have read the responses by authors and am generally happy about the revision. I do notice a new thing in Figure R5. It seems there are more than one redox reaction, different from Figure 2C or 3B, please explain what they are. Once this new question is taken care of, I am recommending the manuscript to be accepted.

Our response: We first appreciate the reviewer's support on our manuscript publication. Further, we have made explanations on the reviewer's concern as below.

To answer the question, please let us lay out the testing parameters first. The CV measurement in Figure 2C was on performed on a three-electrode cell (noted as Al|Mo/In) using Mo wire (very limited surface area) as WE. The two measurements in Figure 3B and Figure R5 were performed on an Al|ACC/In cell using activated carbon cloth (high surface area) as WE, but using different scan rates (0.2 mV/s in Figure 3B, and 1-10 mV/s in Figure R5). Therefore, the three measurements were different in terms of the WE surface area and scan rates; the explanation is detailed below.

As the reviewer reminds, we observe that the Al|ACC/In cell shows a small peak near the equilibrium potential (around 1.1 V) in Figure R5. As the scan rate increase from 1 to 10 mV s⁻¹, this peak becomes more obvious. This phenomenon can be attributed to the difference of reaction kinetics at high scan rates, causing insufficient reaction processes. At high scan rates, the size of the diffusion layer becomes small, and the electrochemical reaction process is diffusion-controlled. In detail, during the initial charging process, the solution-to-solid conversion reaction occurs in a small diffusion layer and is very fast with very small overpotential (occurring around 1.1V). This initiates the deposition of the generated solid InCl₃ on the current collector. In the subsequent charging process, the reaction kinetics is diffusion-controlled, causing a

larger cell voltage polarization. This peak splitting phenomenon was also observed in other battery systems (Nature Energy, 2017, 2, 17090; Electrochimica Acta, 2020, 330, 135314; Chemical Engineering Journal; 2023, 452, 139311.). However, at a low scan rate (in Figure 3B), the Al|ACC/In cell has sufficient reaction processes, resulting in a pair of well-defined redox peaks (Figure 3B).

In Figure 2C, the molybdenum wire was employed as the working electrode in order to investigate of the nature of the indium electrochemistry using a three-electrode beaker-type cell. Due to the very small reaction area of molybdenum wire and relatively high operation temperature, the reaction processes are still sufficient at the scan rate of 2 mV/s, and thus only one pair of redox peaks are observed in Figure 2C.